# The Nurse’s Role in Educating Pediatric Patients on Correct Inhaler Technique: An Interventional Study

**DOI:** 10.3390/ijerph19074405

**Published:** 2022-04-06

**Authors:** Eva Benito-Ruiz, Raquel Sánchez-Recio, Roberto Alijarde-Lorente, Isabel Iguacel, María Pérez-Corral, Carlos Luis Martín de Vicente, Ainhoa Jiménez-Olmos, Ángel Gasch-Gallén

**Affiliations:** 1Pediatric Intensitive Care Unit, Miguel Servet Hospital, 50009 Zaragoza, Spain; evabenitor@unizar.es (E.B.-R.); mariafg180@gmail.com (M.P.-C.); ajimenezolmos@gmail.com (A.J.-O.); 2Physiatry and Nursing Department, Faculty of Health Sciences, University of Zaragoza, 50009 Zaragoza, Spain; rsanchez@unizar.es (R.S.-R.); angelgasch@unizar.es (Á.G.-G.); 3Pediatric Unit, Obispo Polanco Hospital, 44002 Teruel, Spain; ralijarde@yahoo.es; 4Pediatric Pulmonology Unit, University Miguel Servet Hospital, 50009 Zaragoza, Spain; carl_zaragoza@yahoo.es

**Keywords:** early intervention, educational, education, nursing, inhalers, medication errors, pediatrics, respiration disorders

## Abstract

The prevalence of pediatric respiratory diseases in Spain is 23%. Inhalation is the preferred route of administration but there are frequent errors in the performance of the inhalation technique leading a poor control of the disease. The aim of this research was to detect errors in the execution of the inhalation technique at a Pediatric Pulmonology Unit in a hospital of Aragón (Spain). In order to improve the administration of inhaled medication, an educational intervention for 1 year by nursing was conducted. This interventional study, including children aged 1 to 15 years with an inhalation therapy and who attended the Pediatric Pulmonology Unit, was conducted between September 2017 and September 2018. Logistic Regression models were conducted in SPSS. This study involved 393 children (61.1% boys). Before the intervention, 39.4% achieved a correct inhalation technique increasing up to 62.1% after the intervention. Those who had their first visit to the Unit, young children and girls had a higher risk of incorrect performance than those with subsequent visits, older children, and boys, respectively. The most common errors in the inhalation technique were not performing adequate apnoea after inhaling and not rinsing the mouth at the end of the procedure. The education given by nurses to pediatric patients improved the inhalation technique, achieving better control of the disease and use of the health system.

## 1. Introduction

Pediatric respiratory diseases constitute a significant health problem, with prevalence in Spain standing at 23% [1]. At least 50% of children under 6 years of age have experienced wheezing episodes in their lives [2,3]. Asthma attacks are considered one of the main medical emergencies in pediatrics and account for about 5% of the reasons for emergency department visits. About 15% of patients require hospital admission and it is estimated that exacerbations are responsible for more than 80% of the direct costs associated with asthma and emergency department visits [2].

Inhalation is the administration route of choice for adequate management and control. It is essential to achieve a correct approach in patients and their families. The role of nursing is key to fostering appropriate use of the various inhalation devices and to identify risk patients and provide better care and education for the control of chronic respiratory diseases [3,4,5]. Due to the shortage of medical professionals in primary care and the development of specialist nursing training programs, nurses have acquired autonomy in the management of chronic pediatric pathologies. Nurses, in view of their position of proximity, accessibility and trust, and the close and continuous follow-up that they provide, are best placed to become involved in the education of the child and their family. To discharge this primary role as provider of care plans and intervention in chronic illnesses, they must be motivated and trained, in addition to having the necessary time and resources [6,7].

It is necessary to achieve an adequate compliance with treatment, which must be reviewed regularly and any deficiencies improved through proper inhaler management [3,4,5,8]. It has been verified that nurse-led education interventions, in which nurses conduct an assessment of the inhalation technique, which they then follow up on and review, ensure the correct treatment, monitoring and prevention of exacerbations [5,7].

Although studies conducted on the pediatric population reflect more satisfactory results than those conducted on the adult population, there are frequent errors, such as incorrect inhalation, failure to hold the breath for 10 s and not breathing out slowly in the exhalation phase [8], mainly in subjects aged between 2 and 15 [9], leading to uncontrolled asthma. The factors which mostly affect the proper use of the inhalation technique are age, gender, education level, device used and professional review of the inhalation technique [10]. In Spain, most of the studies have evaluated inhaled therapy in adults but not in children [11,12]. The most frequent errors found were failure to maintain apnoea 10 s after inhalation or absence of prior expiration. González Martínez et al. [11] pointed out the importance of educational nursing interventions, showing their usefulness in improving quality of life and the self-control of the disease.

The correct use of inhaled therapy in the pediatric population with respiratory pathology is an understudied topic. Children represent a group at special risk of morbidity and mortality. Correct use of inhalers has been shown to be critical in controlling the disease as well as avoiding associated complications. Nurses, as primary caregivers, can play a key role in educating patients and their families to optimize disease control [5,7].

Therefore, the purpose of this investigation was to detect errors in the execution of the inhalation technique in a Pediatric Pulmonology Unit of a tertiary hospital in Aragón (Spain). In order to improve the administration of inhaled medication in children with respiratory diseases, an educational intervention by nursing professionals was conducted.

## 2. Materials and Methods

### 2.1. Participants

An intervention study at the Pediatric Pulmonology Unit in a hospital of Aragón (an autonomous region in the northeast of Spain) was conducted between September 2017 and September 2018. A total of 393 children (61.1% of whom were boys), aged between 1 and 15 years of age, who attended the Pediatric Pulmonology Unit and used any of the main devices used in the pediatric population as inhalation therapy based on the various clinical practice guidelines namely (inhalation chamber with mask; inhalation chamber with mouthpiece; Accuhaler^®^; Turbuhaler^®^; and Novolizer^®^ [1,4,13] and who agreed to participate in the study were included in the study. Those who refused to participate in the study, patients who did not properly understand the Spanish language, or who had cognitive or psychiatric problems or serious disorders and were incapable of performing the inhalation technique were excluded.

### 2.2. Procedure

The subjects who participated in the study were referred through Primary Care by their referring pediatricians, as well as from Specialized Care after admission to the hospital itself.

Patients or caregivers who used an inhaler device to manage their respiratory disease were asked to demonstrate (with a placebo device) “exactly as they would use it at home”. In order to assess whether the inhalation technique was performed correctly, a checklist was prepared detailing the steps to be taken for each of the devices under analysis, according to the recommendations of the Airways Group of the Spanish Association for Primary Pediatric Care [4]. The inhalation technique was evaluated by a trained investigator physician or nurse who recorded the findings. The inhalation technique was considered correct when all essential steps were properly performed. After the demonstration face to face, correct use of the device was shown to the caregivers and they were given recommendations. During an interview with a nurse, the following information was collected: sociodemographic information such as the age and gender of the patients; pathology for which the patient was attending the pulmonology pediatric department; the treatment/s they were taking; whether it was the first time visiting the pulmonology pediatric department; if they had needed to visit emergency service due to a respiratory disease in the last year; if the children had severe attacks during the last few months; and we also asked about the degree of control afforded by the medicine, which was collected by asking parents about their children’s daytime and night-time symptoms, exacerbations, relief medication and limitation of activity.

The sample selection technique was opportunistic. All the families who attended the specialized consultation were invited to participate in the study, and it was explained to them verbally and in writing what was going to be done, and in the following consultation, if they accepted, they signed the informed consent and the authorization to participate in the study. After inviting the families and/or patients to demonstrate the technique, an education intervention was performed where the nurse, after observing the process, rectified any incorrect steps and performed the technique again in the proper manner. Doubts were clarified and written recommendations were provided regarding the technique. At the next patient check-up, 6 months later, the process was repeated, and the technique was assessed using the same interview as described before.

The guide used based on the design to guarantee the quality of this manuscript was STROBE, obtaining a score of 19 points [14].

All the families who participated in the study agreed to participate and signed the pertinent informed consent form. Permission was requested from the Research Ethics Committee of the Autonomous Region of Aragón (CEICA) (No. C.P.-C.I.PI17/0169-Date accepted: 24 May 2017) and from the Miguel Servet University Hospital. The patients were encoded in the data compilation process to guarantee their anonymity and confidentiality.

### 2.3. Variables

#### 2.3.1. Dependent Variable (Pre- and pOst-Intervention)

The performance in the inhalation technique (pre- and post-intervention) was considered as the dependent variable. The technique was recorded as incorrect if a single error was committed in any of the steps under analysis.

#### 2.3.2. Independent Variable

Sociodemographic variables and clinical variables related to the patient’s pathology were taken into account as independent variables.

Sociodemographic variables such as age and gender (boy/girl) variables were considered as independent variables.

Variables related to the patient’s pathology: different variables that could affect the proper performance of the inhalation technique and the progression of the disease were included as independent variables, namely: the degree of control of the disease; exacerbations; first visit to the pediatric pulmonology clinic (yes/no); visits to the emergency department (yes/no) and medical treatment prescribed.

The degree of control of the disease (controlled vs. partially controlled or uncontrolled [13] was classified with a clinical interview conducted by a pulmonologist on the basis of the items of the Asthma Control Test (ACT) [1]. The ACT provides a score that may help pneumologists to determine whether the treatment plan in asthmatic children, is working or the treatment should be changed. The ACT is a valid and a reliable tool [15,16]. It evaluates the last 4 weeks with the following items: presence and frequency of daytime or nocturnal symptoms; the frequency of use of rescue medication for relief of those symptoms; the maintenance of lung function within or near normal limits; the absence of limitations in daily life, including family, social, work or school activity and physical exercise; and, finally, meeting patient and family expectations regarding the care they receive. A score was given to each of them. The higher the score, the better disease control. In order to facilitate comparison between the groups, the variable was split into two categories (well controlled and poorly controlled). The ACT has been validated and culturally adapted. The ACT has a more detailed validation for use in daily clinical practice, with defined cut-off points, such that a score of 20 or above is highly consistent with well-controlled asthma, a score under 20 was defined as uncontrolled asthma.

Exacerbations (yes/no) correspond with the attacks occurring in the study period, relating to respiratory signs and symptoms and disruption to the work of breathing that alters the patient’s appearance and behaviour [17].

First visit to the pediatric pulmonology clinic refers to whether the patient had previously the Pediatric Pulmonology Unit in a hospital of Aragón (yes/no).

We differentiated between the exacerbations and the number of visits to the emergency department registered in the previous year before the intervention and the exacerbations and the number of visits to the emergency department registered after the intervention (September 2017–September 2018).

### 2.4. Statistical Analysis

In relation to the statistical analyses carried out, we first performed a descriptive analysis of all the study variables. The frequencies and percentages of the categorical variables were calculated, and for the continuous variables, the mean (M) and standard deviation (SD) were calculated. To assess the differences in the following variables—gender, age, first visit vs. no, degree of the control of the disease, exacerbations, emergency department visits, type of inhalation technique) and an uncorrected inhalation technique before and after the educational intervention—we used Chi-squared test for the categorical variables. Lastly, multinomial logistic regression analyses were conducted to identify the abovementioned predictors (and the incorrect performance in the inhalation technique. Raw models and adjusted models by sex and age were conducted.

A statistical significance level of *p* < 0.05 was required in all the tests. All the analyses were conducted in the Microsoft Statistical Package for the Social Sciences program, version 22.0 (SPSS 22).

## 3. Results


**
*Risk factors of the failure in the performance of the inhalation technique before and after the education intervention*
**


A total of 393 boys and girls aged 1 to 15 years old participated in the study. The average age of the participants was 6.3 years old (SD: 3.3). Table 1 shows the risk factors of the failure in the performance of the inhalation technique in children with respiratory disease before and after the education intervention. Those who were between 10 and 15 years old had a higher percentage of making mistakes (66.7%) than younger children (62.0% of mistakes for those who were 1–4 years and 54.4% for those who were 5–9 years). After the intervention, the youngest (1–4 years) were the ones who had a higher risk of making mistakes (44.6% vs. 31.7%). Both before and after the intervention girls had a significantly higher percentage of making mistakes compared to boys (70.1% vs. 54.2% and 47.8% vs. 31.4%, respectively). Appendix A shows the description of the variables included in the study based on the inhalation device used. A direct relationship between age and device used was found. Indeed, the inhalation chamber with mask was used mainly by children aged under 3 years old (M—2.6 years old; SD—1.6), and the average age of users of the inhalation chamber with mouthpiece was 7–8 years old (M—7.6 years old; SD—2.5). The most common errors detected in the inhalation technique were not performing adequate apnoea after inhaling and not rinsing the mouth at the end of the procedure. Asthma (61.6%) and recurrent wheezing (38.4%) were the most prevalent pathologies. It was observed that recurrent wheezing was the predominant pathology among patients using the inhalation chamber with mask (91.0%), with asthma being the most prevalent in respect of the other devices. In total, 61.0% of the patients who attended the consultation for the first time made more errors compared to those who attended routinely. The mean rate of exacerbations in the study period was 1.84 (SD—1.26).

Those who did not have exacerbations had a higher percentage of making mistakes in both pre- and post-intervention (78.0% and 47.5%, respectively) compared to these who had exacerbations (57.9% and 34.7%, respectively). The mean rate of visits to hospital pediatric emergency departments was 0.65 (SD—0.92). Moreover, those who needed to attend the emergency care unit due to a respiratory pathology had a lower percentage of an incorrect performance pre-intervention compared to those who did not attend the emergency unit (54.0% vs. 65.1%) and a higher percentage of an incorrect performance post-intervention (44.9% vs. 35.6%).

Patients categorized as poorly controlled committed a greater number of errors both in the initial test (65.9%) and after the intervention (57.6%) compared to those who were well controlled (59.1% and 32.5%, respectively (Table 1).


**Percentage of errors in the inhalation technique by inhalation device before and after the education intervention**


The initial failure rate was 60.6%. This percentage improved after the education intervention by almost 23%, with better results being observed with the Novolizer^®^ technique (50% of failure rate before the intervention vs. 16.7% after the intervention) (Table 2).

Considering the steps of each technique, in the case of the inhalation chamber with mask device, just 48.6% initially performed the “repeat the procedure” step correctly. Following the intervention, the percentage increased to 91%, with statistically significant differences being found (*p* < 0.001). Regarding the inhalation chamber with mouthpiece, better results were achieved following the intervention in the “Inhale slowly, hold your breath for 10 s and exhale” steps (65.5% vs. 91.4%; *p* < 0.001) and in the “shake and repeat the steps after 30 s if a further dose is required” step (59.8% vs. 75.3%; *p* < 0.001). We observed an improvement in the “Remove the inhaler from your mouth, hold your breath for 10 s and slowly breathe out” step (73.3% vs. 100.0%; *p* < 0.008) and (68.2% vs. 90.9%; *p* < 0.001), with the Accuhaler^®^ and Turbuhaler^®^ devices, respectively (Appendix A).


**Predictors of incorrect inhalation technique in children with respiratory diseases before and after the education intervention**


Lastly, multinominal logistic regressions were conducted to study the factors that influenced the proper performance of the inhalation technique (Table 3). Girls had an increased risk of incorrect performance compared to boys before and after the educational intervention (OR—1.98 (1.29–3.03) and OR—1.98 (1.32–3.04)). Children aged 5 to 9 years old had a lower risk of incorrect performance (OR—0.57 (0.36–0.98)) compared to younger and older children. However, after the intervention, the youngest had a greater likelihood of incorrect performance (OR—1.74 (1.03–2.93)). Those who had a first visit had a higher risk of incorrect performance (OR—2.21 (1.10–4.42)) compared to those who already had a first visit. Moreover, those who had exacerbations before the educational intervention were at lower increased risk for an incorrect inhalation technique (OR—0.34 (0.17–0.65)) than those without exacerbations. Besides, those who had a poor control of the disease had a higher risk of incorrect performance after the educational intervention (OR—2.49 (1.09–5.71)).

No other statistically significant associations were found with other variables such as the type of the inhalation technique and the proper performance of the inhalation technique.

## 4. Discussion

Overall, this study has shown that a high percentage of children with respiratory pathologies in a hospital of Aragon (Spain) do not use their inhalers properly. In particular, more than half of these children made critical errors, most commonly inadequate performing of the apnoea after inhaling and not rinsing the mouth at the end of the procedure. After an educational intervention conducted by nurses, the prevalence of incorrect inhalation technique was significantly lowered. The education given by nurses to pediatric patients improved the inhalation technique, achieving better control of the disease and use of the health system. These results confirmed that educational interventions performed by a pediatric nurse following demonstrations and face-to-face training sessions help achieve quality standards in inhalation therapy and, in turn, support the preliminary work performed at other care levels, such as primary care pediatric consultations [18]. The meta-analysis performed by Kuethe et al., recorded in the Cochrane database, confirmed that nurses could properly take on the management and review of patients with well-controlled asthma [19]. A systematic review conducted by Marufu et al. [20], about nursing interventions to reduce medication errors in pediatrics and neonates, concluded that all interventions demonstrated a reduction in medication errors but most studies performed bundle as opposite to single interventions.

There are studies which demonstrated that age was one of the significant factors for the correct use of the device [21,22]. The best results pre-intervention were achieved in children between 5 and 9 years old, who used mainly the holding chamber and mouthpiece device. Several explanations can be found for this, mainly relating to the intervention of an adult family member who correctly performs the technique [23,24]. One of the reasons for the higher frequency of correct use and adherence to therapy in younger patients in our study may be that treatment is usually administered by parents for younger patients. In the previous analysis, teenagers were those who made the greatest number of errors using the chamber with mouthpiece device and Turbuhaler^®^, respectively. These are the users who benefited the most from the educational intervention carried out, since they obtained an important improvement in the subsequent analysis. Although the National Asthma Guidelines recommended the use of MDI in children over age of 6, the study conducted by Deerojanawong et al. [24] showed that the correct use was significantly higher in children over 10 years old. De Boek et al. [25] observed that children aged 8 and over are more capable of using inhalers and make fewer mistakes. In this study, it can be seen how the older the child is, the lower the risk that they will make mistakes when performing the technique.

In the analysis of errors by inhalation technique, it has been observed that the most common errors were “Hold your breath for 10 s” and “Rinse your mouth when finished and store the inhaler in a dry place”, somewhat similar to what had been described by Weserik et al. and Manriquez et al. in their articles [26,27]. A study conducted by Alexander et al. [28] found that 75% of pediatric patients who were confident in their inhaler technique missed more than four steps. This study underscored the need for inhaler use re-assessment and re-education, despite a patient’s confidence in their ability to use the device. This suggests that these aspects should be emphasized and make sure that these patients improve their technique in future check-ups. It has been reported that most children and adults reduce their mistakes in inhalation device use after receiving education for it. Repeated education results in better inhalation technique and plays a key role in successful inhalation therapy [29].

The results of the present research showed that there was a lower risk of boys incorrectly performing the technique. Other studies did not, however, find statistically significant differences. Further research is required in order to identify the possible impact of gender roles in children at such an early age and how health education interventions may benefit boys more than girls [30], which may affect health results in interventions of this kind [31].

Health education in pediatric patients with respiratory pathologies, when performed and re-assessed on a regular basis, promotes proper use of inhalers [13,32]. Countless studies support the role of nursing in the management of chronic diseases, both in pediatric and adult patients. Laurant et al. [33] describe in their study that around half of patients preferred nursing care when it comes to routine and educational care or support, and Rance et al. [34] affirmed that specialist nurses have proven themselves to be efficient care consultants, capable of being educators and of following up on children with high-risk asthmatic pathologies.

Nurses play a fundamental role in educating children in inhaler treatment, as can be seen in various studies that show a reduction in readmissions among hospitalised children who received training from nursing personnel [24,35]. Regular intervention by actively checking the inhalation technique constitutes an indispensable pillar of healthcare for patients undergoing treatment using inhalation devices, and nursing personnel are key in this process. Kamps et al. [32] also suggested that three repeated instruction sessions were needed to achieve the correct inhalation technique in newly referred patients. Preliminary studies highlight the fact that, in addition to written information [26,36], supplementary support interventions are required in order to review the technique and obtain feedback from the patient and caregivers [37], as has been done in this study.

In addition, visits to the emergency department during the study period were linked to fewer errors in inhalation technique. Scarfone et al. [22] and Muñoz et al. [35] affirmed that the ongoing education intervention performed by healthcare professionals in emergency departments reinforces a proper inhalation device technique and reduces demand in relation to this pathology.

Incorrect use of the device negatively affects the control of respiratory diseases. Insufficient drug delivery may lead to poor drug efficacy and inadequate control and it has been shown that using inhaler devices correctly improves diseases control in pediatrics [23]. Accordingly, a higher frequency of controlled asthma was found in patients using their device correctly. The study by Lasmar et al. [38], performed in a pediatric population under 12 years of age, found higher controlled asthma in patents using medications regularly than the ones using them irregularly.

After conducting this study, it can be affirmed that designing user-friendly inhalation devices and providing recurrent instruction on the management of the devices is essential in order to improve results in the various respiratory pathologies and adherence to treatment. Significant factors to be considered are selecting a preferred device for the patient and identifying whether or not they are capable of using it correctly in order to properly administer the drug [39].

Future lines of research may include the study of the suitability of different devices in children in the control of the disease (especially in those places where the population has more complicated access to hospitals). For this reason, it is necessary to include a greater number of children and different hospitals and health centres.

This study is not without limitations. One of the limitations centers on the relatively low number of participants from a single hospital and the information bias arising from those participating in the study. Attempts have been made to control for information bias by verifying whether the technique was performed correctly. Despite all of the above, the present study shows an improvement in inhaler technique in pediatric patients after the educational intervention, confirming the importance of this type of strategy as an element in the control of respiratory disease in a particularly vulnerable population.

## 5. Conclusions

Chronic respiratory pathologies in pediatrics represent a significant number of visits to health services as well as high costs in the health system. Inhalers are used as one of the most effective treatments in these pathologies. This study demonstrated that the inhalation technique was incorrect for many users, especially for children who had their first visit to the unit, young children and girls, compared to those with subsequent visits, older children, and boys, respectively.

In the present study, an educational intervention led by nurses improved the inhalation technique. The most common errors in the inhalation technique were not performing adequate apnoea after inhaling and not rinsing the mouth at the end of the procedure.

Nursing professionals as front-line caregivers and patient educators must take a position and play an essential role in following up on pediatric patients with respiratory pathologies by performing education and supporting interventions on the correct use of inhalation devices and subsequently assessing the feedback obtained from the patient.

### Implications for Research, Policy, and Practice

This research provides novel information on the management of inhalation devices in pediatric population with respiratory disease in the hospital setting, a field that has been little investigated. These children represent a group at special risk of morbidity and mortality.

Inhaler competency is an integral component of effective self-management in respiratory diseases, and nurses play a pivotal role in delivering education to the patient and caregiver that is required to optimize disease control. Educational interventions such as the one shown in this research are necessary, where education led by nurses based on direct observation of the technique and feedback from the patients is essential.

## Figures and Tables

**Table 1 ijerph-19-04405-t001:** Risk factors of the failure in the performance of the inhalation technique in children with respiratory disease before and after the education intervention. Results from the Crosstabs using Chi-square test (significant differences were considered at the 0.05 significant level).

Variables		Incorrect Performance Pre-Intervention % (N)	*p*-Value	Incorrect Performance Post-Intervention % (N)	*p*-Value
Gender	Girl	70.1 (110)	0.002	47.8 (75)	0.001
	Boy	54.2 (128)		31.4 (74)	
Age	1–4 y	62.0 (75)	0.110	44.6 (54)	0.114
	5–9 y	54.4 (81)		37.6 (56)	
	10–15 y	66.7 (82)		31.7 (39)	
First visit ^1^	Yes	61.1 (22)	0.943	55.6 (20)	0.002
	No	60.5 (216)		36.1 (129)	
Degree of control of the disease (pre-intervention) ^2^	Good	59.1 (182)	0.257	34.7 (107)	0.014
	Poor	65.9 (56)		49.4 (42)	
Degree of control of the disease (post-intervention) ^2^	Good	62.0 (191)	0.262	32.5 (100)	0.000
	Poor	55.3 (47)		57.6 (49)	
Exacerbations (pre-intervention) ^3^	Yes	57.5 (192)	0.003	36.2 (121)	0.101
	No	78.0 (46)		47.5 (28)	
Exacerbations (post-intervention) ^3^	Yes	58.6 (164)	0.204	36.1 (101)	0.236
	No	65.5 (74)		42.5 (48)	
Emergency care (pre-intervention) ^4^	Yes	54.0 (87)	0.028	37.3 (60)	0.826
	No	65.1 (151)		38.4 (84)	
Emergency care (post-intervention) ^4^	Yes	67.3 (66)	0.113	44.9 (44)	0.100
	No	58.3 (172)		35.6 (105)	

y—years old. ^1^. First Visit—if the patient had visited for the first time the Pediatric Pulmonology Department in September 2017. ^2^. Degree of control of the disease—classified based on the clinical interview and the following items: daytime and night-time symptoms, exacerbations, relief medication, limitation of activity, number of exacerbations (number of activities during the study period). ^3^. Exacerbations—relapses due to the respiratory disease during the study period. ^4^. Emergency Care—if they visited the hospital emergency department due to the respiratory disease during the study period.

**Table 2 ijerph-19-04405-t002:** Failure rate in the inhalation technique before and after the education intervention, based on the various inhalation devices in the pediatric population with respiratory pathologies.

Inhalation Device	Pre-Test % (N)	*p*-Value	Post-Test % (N)	*p*-Value
Accuhaler^®^	66.7 (20)	0.696	40.0 (12)	0.361
Turbuhaler^®^	54.4 (36)		31.8 (21)	
Novolizer^®^	50.0 (6)		16.7 (2)	
Chamber with mask	61.3 (68)		42.3 (47)	
Chamber with mouthpiece	62.1 (108)		38.5 (67)	

**Table 3 ijerph-19-04405-t003:** Predictors of incorrect inhalation technique in children with respiratory diseases before and after the education intervention (reference: correct inhaltaion technique). Results from the logistic regression models, showing both odds ratio (OR) and confidence intervals (CI 95%) in a raw model and adjusted model (age and sex).

	Incorrect Inhalation Technique			
	Raw Models		Adjusted Models	
	Pre-Intervention	Post-Intervention	Pre-Intervention	Post-Intervention
Variables				
Gender				
Girl	**1.98 (1.29–3.03)**	**2.00 (1.32–3.04)**		
Boy	Ref.	Ref.		
Age				
1–4 y	0.82 (0.49–1.38)	**1.74 (1.03–2.93)**		
5–9 y	**0.57 (0.36–0.98)**	1.30 (0.78–2.15)		
10–15 y	Ref	Ref.		
**Pre-intervention**				
First visit ^1^				
Yes	1.03 (0.51–2.07)	**2.21 (1.10–4.42)**	0.88 (0.27–2.81)	1.78 (0.51–6.11)
No	Ref.	Ref.	Ref.	Ref.
Control of the disease ^2^				
Good	Ref.	Ref.	Ref.	Ref.
Poor	1.34 (0.81–2.21)	0.76 (0.46–1.23)	1.36 (0.62–2.95)	0.66 (0.34–1.28)
Exacerbations ^3^				
Yes	**0.38 (0.20–0.74)**	0.74 (0.47–1.17)	**0.34 (0.17–0.65)**	0.63 (0.27–1.45)
No	Ref.	Ref.	Ref.	Ref.
Emergency care ^4^				
Yes	**0.63 (0.42–0.95)**	0.58 (0.30–1.59)	0.95 (0.63–1.44)	0.96 (0.47–1.96)
No	Ref.	Ref.	Ref.	Ref.
**Post-intervention**				
Control of the disease ^2^				
Good	Ref.	Ref.	Ref.	Ref.
Poor	0.76 (0.46–1.22)	**2.83 (1.73–4.64)**	0.66 (0.34–1.28)	**2.49 (1.09–5.71)**
Exacerbations ^3^				
Yes	0.75 (0.47–1.17)	0.72 (0.44–1.17)	0.76 (0.49–1.19)	0.66 (0.35–1.26)
No	Ref.	Ref.	Ref.	Ref.
Emergency care ^4^				
Yes	**1.59 (1.05–2.40)**	1.27 (0.65–2.45)	1.47 (0.92–2.35)	1.19 (0.46–3.07)
No	Ref.	Ref.	Ref.	Ref.
Inhalation technique				
Accuhaler^®^	1.22 (0.54–2.78)	1.07 (0.48–2.54)	0.64 (0.15–2.60)	1.33 (0.42–4.13)
Turbuhaler^®^	0.73 (0.41–1.30)	0.75 (0.41–1.36)	0.30 (0.09–1.03)	0.91 (0.35–2.39)
Novolizer^®^	0.61 (0.19–1.98)	0.32 (0.07–1.51)	0.21 (0.02–1.57)	0.38 (0.04–3.13)
Chamber with mask	0.97 (0.59–1.58)	1.17 (0.72–1.91)	1.01 (0.34–2.98)	0.90 (0.36–2.25)
Chamber with mouthpiece	Ref.	Ref.	Ref.	Ref.

In **bold** significant results; ^1^. First Visit—if it is their first visit to the Pediatric Pulmonology Department. ^2^. Degree of control of the disease—classified based on the clinical interview and the following items: daytime and night-time symptoms, exacerbations, relief medication, limitation of activity, number of exacerbations (number of activities during the study period). ^3^. Exacerbations—relapses due to the respiratory disease during the study period. ^4^. Emergency Care—if they visited the hospital emergency department due to the respiratory disease during the study period.

## Data Availability

The raw data supporting the conclusions of this article will be made available by the authors upon request, without undue reservation.

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
