# Peer review of "The Nurse’s Role in Educating Pediatric Patients on Correct Inhaler Technique: An Interventional Study"

_ijerph, 2022, doi:10.3390/ijerph19074405_

Round 1

Reviewer 1 Report

Thank you for submitting the manuscript to this journal. You have chosen an interesting subject to do this study. Some issues need to be clarified. I have summarized these concerns below: the following comments need to be taken with care in order to improve the quality of the manuscript for publishing.

TITLE

In the title it would be appropriate for the authors to indicate the design of the study. The term perspective can lead to confusion as to whether it is a qualitative or quantitative study.

ABSTRACT

The authors use the first person plural, change to impersonal forms.

To review the general objective of the study, the authors propose two objectives and even address the term intervention, and in the design they state that it is a descriptive study. Therefore, it is necessary for the authors to review this and clarify this issue in the manuscript.

Line 33: 393 children… Modify wording so that the sentence does not start with a number.

KEYWORDS

keywords must be in alphabetical order. Likewise, within the key words pediatrics should be included.

INTRODUCTION

Line 40. the authors do not provide reference for the exposed idea. It is important that the entire introduction is referenced and based on the most current references possible.

 It would be interesting if the authors include a paragraph with the studies carried out in Spain on this subject. Likewise, it would be interesting to highlight the interest and the need to carry out this study, since it is not clear what is new.

The aim must be revised, and include a single general aim.

METHODS

Line 69-103: In the methods section, all subsections should be clearly displayed. The information included at the beginning is not included in a subsection, it would be participants, procedure...

the authors comment that it is a descriptive study and later indicate that they carry out an intervention, could you clarify this?

This section is incomplete the following subsections are missing:

Design and Participant (indicate criteria for inclusion and exclusion of participants, calculation of sample size...).

What is the guide used based on the design to guarantee quality? STROBE, TREND…

How was the data collected to guarantee anonymity, how were the parents contacted? these aspects must be included in the manuscript.

How the authors evaluate the dependent variable is not clear.

They could indicate the reliability and validity of the questionnaire used in that population. Asthma Control Test (ACT).

Exacerbations, First visit to the pediatric pulmonology clinic, this would not be clinical characteristics collected, they could go right after the sociodemographic characteristics.

What type of variable is the intervention? it is not clear from the manuscript.

To assess the effect of the intervention, a comparative analysis was performed between the results of the questionnaire on the correct application of the inhalation technique before and after the invention and related sample comparison techniques. It would be necessary for them to indicate the statistical tests.

RESULTS

The writing of the results is not structured, please include subsections. This might help readers.

Check the wording of the results, information from the text is duplicated in the tables.

They could include an image of the regression performed.

DISCUSSION

Avoid the first person plural in the wording.

Please check the wording of the discussion, avoid duplicating the results in this section. Encourage discussion by comparing with other previous manuscripts.

The beginning of the discussion appears in bold, unifying the style.

No numerical data in the discussion.

It would be necessary to include new future lines to investigate, after the completion of the study.

CONCLUSION

The conclusions must be clear, direct and respond to the stated objective, for this reason the authors must review them.

REFERENCES

References must be ordered and numbered according to their appearance in the text.

Author Response

Dear Editor-in-Chief of International Journal of Environmental Research and Public Health,                                          

Please find enclosed a revised copy of our proposed manuscript that we would like to resubmit to the editorial board of the journal “International Journal of Environmental Research and Public Health” to reconsider it for publication. Enclosed you will find a revision of our manuscript, “The nurse’s role in educating pediatric patients on correct in-haler technique. An intervention study”. We would like to thank the editors and reviewers for their thoughtful and constructive comments. Changes in the manuscript have been highlighted using track changes. An itemized point-by-point response to the editor comments is presented below. 

This manuscript contains material that is original and not previously published in text or on the Internet, nor is it being considered elsewhere until a decision is made as to its acceptability by the International Journal of Environmental Research and Public Health Editorial Review Board.

Dr. Isabel Iguacel Azorín

On behalf of all the co-authors.

*Correspondence to: Isabel Iguacel

Reviewer 2 Report

-Similar studies have already been carried out. Therefore, this study does not have a novel overtone.
-
The text requires linguistic correction.
-The data in table 2 are also 
presented in table 1, what is more, there are some discrepancies between these data.
- p values ​​should be given in the tables.
-'The higher the score, the lower disease control' - line 127, and then 'score of 20 or above is highly consistent with well-controlled asthma, a score under 20 was defined as uncontrolled asthma' - line 131 = these sentences contradict each other.
-'compared to younger and older children and older children' - line 227.

Author Response

(The authors gave the same response as above.)

Round 2

Reviewer 1 Report

Thank you so much. All the recommendations were addressed.